# An evolution-based high-fidelity method of epistasis measurement: Theory and application to influenza

**Gabriele Pedruzzi, Igor M. Rouzine** [ID] *

Sorbonne Université, Institute de Biologie Paris-Seine, Laboratoire de Biologie Computationelle et Quantitative LCQB, Paris, France

* igor.rouzine@sorbonne-universite.fr

**Data Availability Statement:** Influenza sequence data are taken from public database https://www.fludb.org. Our software is deposited at https://github.com/rbatorsky/hiv-recombination, https://github.com/irouzine/Pedruzzi To characterize the

## Abstract

Linkage effects in a multi-locus population strongly influence its evolution. The models based on the traveling wave approach enable us to predict the average speed of evolution and the statistics of phylogeny. However, predicting statistically the evolution of specific sites and pairs of sites in the multi-locus context remains a mathematical challenge. In particular, the effects of epistasis, the interaction of gene regions contributing to phenotype, is difficult to predict theoretically and detect experimentally in sequence data. A large number of false-positive interactions arises from stochastic linkage effects and indirect interactions, which mask true epistatic interactions. Here we develop a proof-of-principle method to filter out false-positive interactions. We start by demonstrating that the averaging of haplotype frequencies over multiple independent populations is necessary but not sufficient for epistatic detection, because it still leaves high numbers of false-positive interactions. To compensate for the residual stochastic noise, we develop a three-way haplotype method isolating true interactions. The fidelity of the method is confirmed analytically and on simulated genetic sequences evolved with a known epistatic network. The method is then applied to a large sequence database of neurominidase protein of influenza A H1N1 obtained from various geographic locations to infer the epistatic network responsible for the difference between the pre-pandemic virus and the pandemic strain of 2009. These results present a simple and reliable technique to measure epistatic interactions of any sign from sequence data.

## Author summary

Interactions between genomic sites create a fitness landscape. The knowledge of topology and strength of interactions is vital for predicting the escape of viruses from drugs and immune response and their passing through fitness valleys. Many efforts have been invested into measuring these interactions from DNA sequence sets. Unfortunately, reproducibility of the results remains low due partly to a very small fraction of interaction pairs and partly to stochastic linkage noise masking true interactions. Here we propose a

three-dimensional network of epistatic interaction in Fig 4, we used software package ChimeraX from internet site https://www.rbvi.ucsf.edu/chimerax/.

**Funding:** This research has been funded by l'Agence Nationale de la Recherche, grant J16R389 to IMR, http://www.agence-nationale-recherche.fr/. The funders had no role in study design, data collection and analysis, decision to publish, or preparation of the manuscript.

**Competing interests:** The authors have declared that no competing interests exist.

method to separate stochastic linkage and indirect interactions from epistatic interactions and apply it to influenza virus sequence data.

## Introduction

About a century ago, it was realized that the evolution of a population is strongly affected by the fact that the fates of alleles at different loci are linked unless separated by recombination. These *linkage* effects include clonal interference [1,2], background selection, genetic hitchhiking [3], enhanced accumulation of deleterious mutations (Muller's ratchet) [4], and the increase of genetic drift at one locus due to selection at another [5]. Linkage decreases the speed of adaptation and creates random associations between pairs of mutations occurring on the same branch of the ancestral tree.

These effects have been taken into account in early mathematical models considering two loci [6] and, more recently, in the traveling wave approach, which describes an arbitrarily large number of linked sites [7–12]. These models describe the dynamics of fitness classes and include the factors of selection, mutation, random genetic drift and recombination [13–16]. All these models predict a narrow fitness distribution traveling in the fitness space in a direction depending on the initial conditions and parameters [17,18]. This "traveling wave" consists of the deterministic bulk and the leading stochastic edge, where the generation and establishment of rare beneficial mutations limit the adaptation rate. Alternatively, the distribution may move backwards accumulating more and more deleterious alleles (Muller's ratchet). These models are able to express, in the general form, important observable quantities in terms of model parameters, such as the population size, mutation rate, and the distribution of selection coefficients over loci. The observable quantities include the adaptation rate [7–10], Muller ratchet rate [7,9], the conditions of full equilibrium [7,19], fixation probability of an allele, and the most probable selection coefficient [12]. The same general approach was used to predict the statistical properties of the ancestral tree [15,16,20–22].

Despite of all the progress, prediction of the evolution of specific sites in the multi-site context remains an open question. How do allelic frequencies at each site change in time when the system is adapting? Although the dependence of allelic frequencies on time is stochastic due to the combined effects of selection, random drift, and linkage, what can be said about the average allelic frequency of a given site with a given fitness effect of mutation? Also, what can we say about the evolution of site pairs, especially in the presence of epistatic interaction?

Epistasis defined as the interaction of genes and gene regions contributing to phenotype is an omnipresent phenomenon [23]. Gene interactions are reported to be responsible for a considerable fraction of the organism's genetic inheritance [24]. They create fitness valleys in the evolutionary path [25]. In pathogens, epistasis facilitates the development of drug resistance and immune escape and impedes reversion of drug-resistant mutations [26–31]. Most of HIV variation in untreated patients has been argued to arise from mutations compensating early immune escape mutations [32].

Pairwise epistasis can be measured from binding free energy [33] and measuring fitness gains [34]. A large number of approaches have been proposed to measure epistasis from genomic data [35–37]. The simplest methods are based on pairwise allelic correlations [5,38]. The problem with all these approaches is that linkage and indirect interactions create strong inter-site associations even between non-interacting locus pairs, and these false-positive pairs are much more numerous than the true epistatic pairs. Stochastic effects are well-recognized as the most serious obstacle to the detection of epistatic effects [39]. In a single asexual

population, stochastic linkage completely overshadows the epistatic footprint, except in a narrow range of times and parameters [40]. The same limitation exists for the tree-based methods of detection [41,42].

A method to eliminate false-positive links arising due to indirect epistatic interactions in the absence of linkage has been developed and successfully applied to protein sequences isolated from different species [43,44]. A similar technique has been applied to the fitness landscape of antibody-binding regions of HIV protein gp120 [45]. However, none of these methods enable reliable measurement of epistasis in asexual populations or in sexual populations at close loci from the same species [39]. Any attempt to detect epistasis, whether by using covariance measures ($D'$, $r^2$, mutual entropy, universal footprint of epistasis (UFE) [46]) or the tree-based methods [42] faces the same problem, the overwhelming linkage effects. The existing methods are based on the approximation of quasi-linkage equilibrium which neglect linkage effects assuming the limit of strong recombination (see [47] for review).

As we have shown in [40], the increasingly dominant effect of linkage over epistasis in allelic associations results from the random divergence of independent populations in time, so that all sequences are similar to their most recent common ancestors, and the common ancestors move away from the origin and from each other along stochastic trajectories [40]. As a result, any measure of co-variance, or even the use of the entire tree, produces only strong noise of random sign. Co-variation due to random linkage completely masks the epistasis signature in a population. The only way to resolve this issue is to average the haplotype frequencies over many independent populations with similar parameters under similar conditions. Without sampling multiple populations, it is not possible to infer epistasis in principle, due to the stochastic nature of phylogenetic relation of sequences. This fundamental limitation resulting from random phylogeny [40] cannot be resolved by any existing or future method. Furthermore, as we show below, even 200 populations may be not enough to eliminate the false-positives in a system of 40 loci. The contribution of the present work is not to overcome this fundamental limitation, which is not possible in principle, but to demonstrate the existence of a large number of residual false-positive interactions left after averaging over an ensemble of 20 to 200 populations, and to propose a new method to eliminate these false links.

The new technique is based on the use of three-way haplotypes. The basic idea behind this method is that demanding a majority allele at a neighbor site of a measured pair of sites interrupts a "detour", i.e., a path along interacting sites that creates a false-positive interaction for the pair of interest. In other terms, the additional condition splits the genome into independent blocks.

The high fidelity of this detection technique is demonstrated below by using two parallel methods: analytic derivation for a simple network topology and Monte-Carlo simulation. The analytic derivation makes no assumptions about the range of $E$ or sign or epistasis. The simulation example considers compensatory epistasis, $0<E<1$, but the technique applies at any value of $E$, including negative epistasis.

Then, we apply the method to real virus sequences from an adapting population. Influenza virus evolving in a human population can be mapped onto the traveling wave theory with an effective selection pressure caused by accumulating memory cells [48,49]. Therefore, it is expected to be amenable to our method. Using more than 8000 influenza sequences obtained from various geographic locations, both before and after the pandemic of 2009, we use our method to predict the epistatic interactions among alleles from their observed associations by isolating them from linkage effects in a surface protein, neurominidase. We chose this specific protein, because it underwent strong changes when it was replaced by a new strain in 2009. The old strain and the pandemic strain of H1N1 share 80% of homology. This indicates that, at some point in the past, the two strains had a common ancestor.

We infer the primary and compensatory mutations that allowed the new strain to outcompete the old strain. The method cannot infer the order in which these compensatory mutations have emerged but only the resulting network. In a similar fashion, epistatic networks in some common proteins are estimated from the comparison between their sequences in long-diverged species neglecting linkage effects [43,44]. The difference in our case is that our approach does not make such an assumption and eliminates linkage effects as well.

Thus, the aim of our theoretical paper is to offer a new method of epistatic detection based on an analytic derivation, test it on simulated sequence data, apply it to a real sequence set, and make a testable prediction for a protein network. We do not aim to describe the evolution of H1N1 influenza strain in detail leaving it for other projects.

## Results

### Simulation model to generate sequences for the test

We start by simulating the evolution of a haploid asexual population using a Wright-Fisher process including the factors of random mutation, random genetic drift, and constant directional selection (Fig 1A) (*Methods*). We assume two possible alleles at each locus (site), 0 and

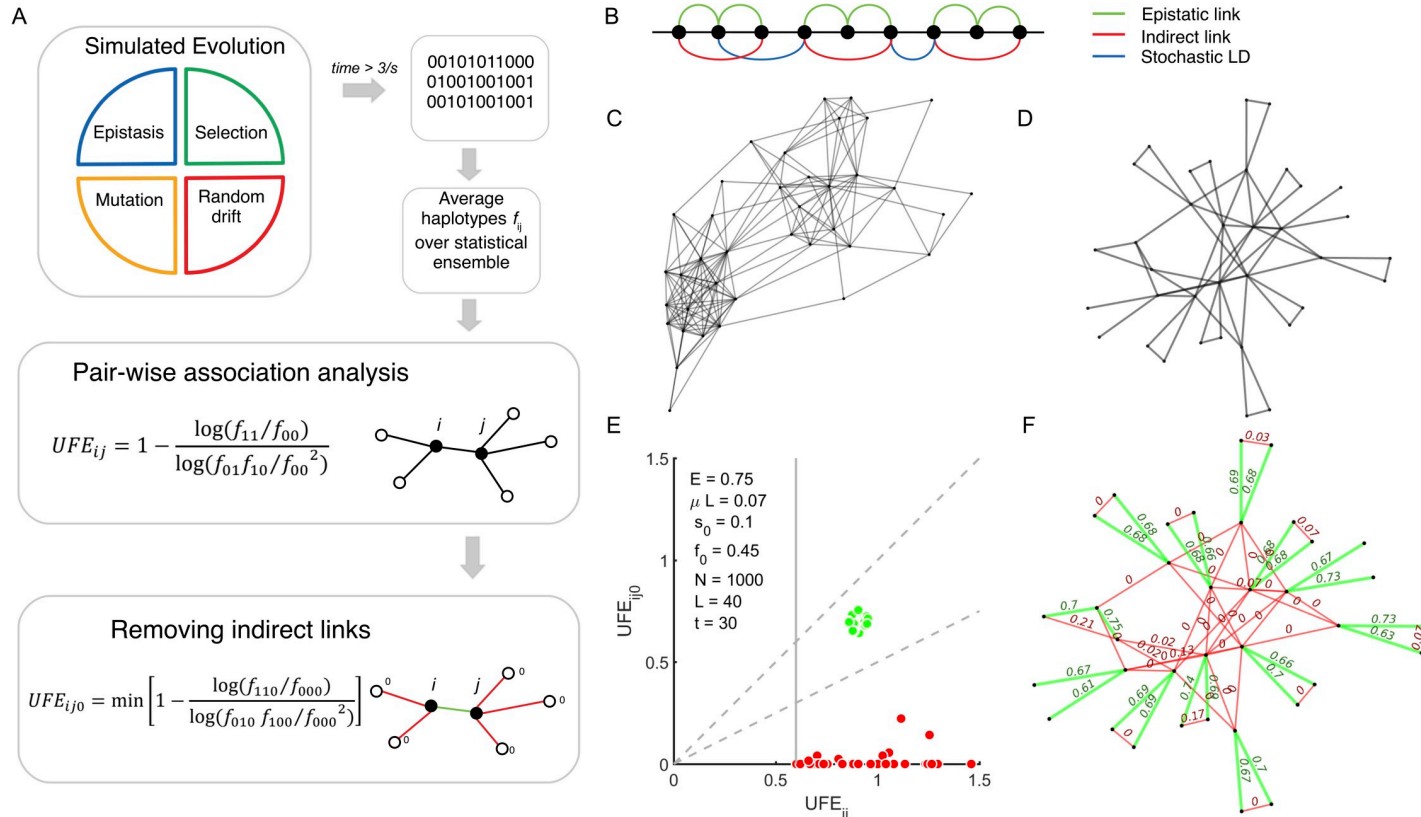

**Fig 1. Schematic diagram of the method and its testing on simulated sequence data. A**. The computer model of asexual evolution includes the factors of random mutation, selection, epistasis, and random genetic drift. Pairwise haplotype frequencies $f_{ij}$ are averaged over simulation runs (independent populations). The pairwise correlation measure $UFE_{ij}$ is calculated from Eq 1. The indirect links and the residual linkage are detected and filtered out by using the tri-way correlation measure, $UFE_{ij0}$, from Eq 2. **B**. Pre-set epistatic network for 50 sites. Green curves: real epistatic interactions. Red lines: indirect interactions. Blue lines: examples of stochastic linkage effects. **C-D**. The network of strong ($UFE_{ij} > 0.5$) candidate epistatic interactions predicted (**C**) from a single population and (**D**) after averaging over 200 populations. **E**. Scatter plot of the three-way haplotype, $min(UFE_{ij0})$ shown against $UFE_{ij}$ for the pairs identified in (**D**). The dashed sector corresponds to the direct interactions. The upper dashed line is the diagonal, $UFE_{ij} = UFE_{ij0}$, and the lower dashed line separates direct and indirect interactions. **F**. Predicted network accurately recapitulates the pre-set epistatic network. Parameters: initial allele frequency $f_0 = 0.45$, mutation rate per genome $\mu L = 0.07$, fixed selection coefficient $s = 0.1$, $N = 1000$, $L = 40$, epistatic strength $E = 0.75$.

1, where 1 stands for an allele that decreases the logarithm of fitness by a fixed value $s \ll 1$ when it does not interact with any other sites. The effects of variable selection coefficients are addressed elsewhere [12,50]. The binary simplification provides a major reduction in the computational cost and is accurate for relatively conserved sequences. We let some pairs of sites to interact epistatically, with a mutual degree of compensation between deleterious alleles chosen to be $E = 0.75$.

We consider a simple epistatic network consisting of double arches (Fig 1B). The network has three types of correlations: direct interactions, indirect interactions, and linkage, as shown by different colors. If the first and the second loci interact, and the second loci and the third loci interact, this leads to correlations between the allelic composition at the first and third loci, even though they do not interact, termed "indirect interaction". In addition, any pair of loci correlates due to linkage, i.e., having a common ancestor. A more complex network is discussed later on.

### First step: Averaging over populations

Genome sequences produced by the simulation demonstrate the presence of strong pairwise correlations between the allelic composition of different loci originating from three sources: direct epistatic interaction, indirect interaction, and stochastic linkage effect. Our first task is to detect all potential epistatic interactions using correlation analysis. As we mentioned, their detection is masked by strong stochastic linkage arising from common ancestors [40]. To decrease linkage effects, for each pair of sites $(i, j)$, we calculate pairwise haplotype frequencies, $f_{\alpha\beta}^{ij}$, where $\alpha, \beta = 0$ or 1, and then we average them over multiple evolutionary-independent populations of the same size. Then, we calculate a metric termed "universal footprint of epistasis" (UFE)

$$UFE_{ij} = 1 - \frac{\log(f_{11}/f_{00})}{\log(f_{01}f_{10}/f_{00}{}^2)} \qquad (1)$$

where $f_{00}, f_{10}, f_{01}, f_{11}$ are the haplotype frequencies averaged over the ensemble of populations (we dropped indices $i, j$). More traditional correlation measures, such as $D'$ and Pearson coefficient $r^2$, have been shown to generate similar stochastic noise [40]. As compared to these measures, $UFE_{ij}$ has the unique advantage of directly measuring the degree of mutual compensation of two alleles for infinite averaging, $UFE_{ij} = E$, provided the interacting pair is epistatically isolated from the other sites [46]. Because the logarithms in Eq 1 diverge when one of the four haplotype frequencies is zero, we consider only site pairs such that all four $f_{ij}$ are larger than $f_{cut}$, where $f_{cut} \ll 1$ is a low cutoff, which is set below at $f_{cut} = 0.05$.

Next, we keep only pairs with sufficiently high correlation, $UFE_{ij} > 0.6$ (we remind that we set $E$ at 0.75). For a single population, the raw graph of inferred pairs is extremely complex and completely hides true epistatic interactions (in this case, 24 interactions) (Fig 1C). For a genome longer than $L = 40$ we took here, it would be even worse. A significant reduction of the number of false-positive interactions is obtained by averaging $f_{ij}$ over 200 independent populations (Fig 1D). However, we can see that the vast majority of remaining links are still false-positive, because their number is still much higher than the number of actual interactions (Fig 1B, green arches).

### Step 2: Three-way correlation

To clean the network from residual false-positive links caused by incomplete averaging over ensemble, we can either try to average over tens of thousands of independent populations, which are never available in real life, or use a trick, as follows.

The idea of the procedure is based on the fact that false-positive links are created by "detours" around pairs of sites, such as chains of indirect epistatic interactions or sites linked due to the common phylogenetic origin (red and blue curves in Fig 1B). To break up the association created by detours, we demand that a neighbor site of the site pair of interest is 0 (better-fit, wild-type) and recalculate the correlation. If that neighbor site happens to be at the most important detour, this condition will break up or, at least, decrease the indirect correlation. Direct interactions are affected by such an additional condition to a smaller extent (if at all). Thus, for each connected pair $i, j$, we calculate the three-way measure

$$UFE_{ij0} = 1 - \frac{\log(f_{110}/f_{000})}{\log(f_{010}f_{100}/f_{000}{}^2)} \qquad (2)$$

where 0 in the third position selects the sequences with the consensus allele 0 at a chosen site adjacent to one site of the tested pair. We consider all possible connected sites, one by one, as the 0-node and calculate the minimum value of $UFE_{ij0}$ over all possible 0-nodes, $\min(UFE_{ij0})$. Finding the minimum not only detects a detour but also finds the most important detour if there is more than one. Thus, we can identify and remove false-positive links as those with a low ratio $\min(UFE_{ij0})/UFE_{ij}$.

For every potential link between sites $i$ and $j$ detected in Fig 1D, we calculate $\min(UFE_{ij0})$ (Fig 1A, bottom). The scatter plot in Fig 1E demonstrates that, for the false-positive pairs, $\min(UFE_{ij0})$ is several-fold smaller than $UFE_{ij}$ (red dots in Fig 1E). For true links (green dots in Fig 1B and 1E), the two correlation measures are nearly the same. The choice of the threshold in $\min(UFE_{ij0})/UFE_{ij}$ (low dashed line in Fig 1E) is not crucial, as long as we average the haplotype frequencies over at least ~20 populations (for our parameter choice). In this case, the two groups, false-positive and true interactions, remain distinct. The end result is 100% perfect detection (Fig 1B). As a bonus, we obtain accurate estimates for the compensation strength: $UFE \approx E$ within 15% accuracy (Fig 1F, numbers at green links).

In our previous paper [40], we showed that averaging over dozens of populations is required to isolated epistatic links. In our example in Fig 1, we demonstrate that, at $L = 50$ sites and moderate population sizes, a half of false-positive links remain even after 200 populations. The present method provides 100% fidelity at $L = 40$ sites or less, for the number of replicate populations between 20 and 200, and epistatic strength $E = 0.5$–$0.75$. Performance drops sharply at $E < 0.25$.

## Analytic results

In the *Methods*, we show analytically that $\min(UFE_{ij0}) \ll \min(UFE_{ij})$ for indirect interactions, as given by

$$UFE_{ij} \approx \frac{1}{4(1-E)}$$

$$UFE_{ij0} \approx 0$$

and that this condition can be used to eliminate indirect interactions analytically, in the general form. In the analytic derivation, we assume that the system is under directed selection and in the multiple-mutation regime (the traveling wave regime), which takes place if $\log(NU) \gg \log(s/U)$ [12]. In this regime, selection sweeps at many sites overlap in time and interfere with each other [7]. We also assume that the population is far from mutation-selection-drift equilibrium, so that deleterious mutation events are negligible. The derivation given in the *Methods* applies at negative values of $E$ as well, but we focus on positive $E$, which case is

termed "diminishing returns epistasis". The reason for this choice is strong effects of epistasis and strong indirect interactions in this region. The analytic derivation applies at any $E < 1$, i.e., below the full compensation point, which covers all basic types of epistasis.

We also repeated this derivation for a more complex topology of closed squares (S1A Fig). Here indirect interaction occurs between the opposite corners of the square, and direct interaction between the sites of one side. This topology is more complex, because it has a loop, and there are two paths connecting the opposite corners. The results show that at $E > 1/3$, the magnitude of direct and indirect correlations is the same. The 3-way correlation method decreases indirect correlation to a larger degree, which can be used to tell the direct and indirect correlations apart (S1B Fig and S1 Table). Thus, the 3-way method is robust with respect to a topology. The difference from the double-arch case is that indirect correlation does not disappear completely and remains of the same order of magnitude as the direct interaction. This is especially true if $E$ is close to the full compensation point (in this case, $E = 1/2$). Because, in real biological systems, the value of $E$ varies broadly across pairs, such a difference may be not enough for the reliable detection of the true links demonstrated above for a loop-less topology. The natural way to address this issue is to add another 0 and measure a 4-way correlation to interrupt both equal detours connecting the two points. This trick, indeed, removes indirect interactions completely in the entire interval of $E$, as given by $UFE_{ind}^{00} \equiv 0$ (S2 Fig and S1 Table).

For the general topology with many loops, the number of the additional zeros required to kill an indirect interaction completely is equal to the number of directions in which a detour can occur (the connectivity parameter). For example, if a chosen site of interest has six neighbor sites with strong correlation (direct or indirect, we do not know), and three of them create a distinct detour to the other site of interest, we would need to add three zeros and calculating the minimum over all possible combination. Therefore, to decipher a very complex network, one needs to add extra zeros iteratively around sites with many neighbors and see if anything has changed at each point. In the example with virus data below, a loopless network emerges already after the three-way test.

## Application to influenza A virus

After testing our method analytically and on simulated sequences, for the sake of demonstration, we now infer an epistatic network for an evolving viral population. Our choice is the surface protein of Influenza A H1N1, neuraminidase (NA), important for virus infectivity and an important target of drug therapy and immune response. This protein is one of two proteins that control the virus entry into a host cell (the other is Hemaglutinin).

Our aim is to identify mutations and their interactions that allowed the pandemic strain of 2009 to outcompete the pre-2009 strain. For this aim, we compared the sequences of the first strain to the sequences of the second strain, both sampled worldwide from dozens of locations. In order to be able to apply our method, we assumed that various local populations are nearly independently evolving, and migration between them is very slow.

Although they are not actually independent, but represents parts of one metapopulation, this is the best approximation one can obtain in any study working with world pandemics data.

We have used about 8000 sequences found in the cited database for the period 2000–2010 from various geographic locations. Our goal was to understand the difference between the strains before and after pandemic of 2009, which have 80% of mutual homology. We wished to infer only the epistatic sub-network related to that difference. For this end, we compared worldwide samples of sequences from the two strains. We randomly sampled similar amounts of sequences from the first and second strains, and re-sampled them several hundred times.

We also checked the robustness of the results to the exact sampling size (Fig 1F). We have observed that the old and the new strains are both diverse. The two strains evolved together over several years, and the new strain gradually replaced the old strain. In terms of travelling wave, that implies that we have two traveling waves with overlapping fitness distribution, one is gradually waning over several years and replacing the other.

To simplify our task, we binarized the sequences by setting each consensus allele to 0 and each non-consensus ("mutant") allele to 1. This simplification is adequate for the aim of detection of interactions, unless several amino acid variants are present at a site at similar frequencies, which we found to be a rare occurrence. We considered only the sites that were strongly polymorphic ($>$5%) and observed a bimodal distribution of sequences in the mutant allele frequency per genome with two separate maxima of different height, at $f = 0.05$ and $0.2$. The low-frequency peak was taller. The bimodal distribution reflects the mixture of two strains, the old and the new, with 80% homology. Thus, the old and the new strain differed in NA in about 100 sites, which is 20% of the length.

In order to compensate for unequal sampling from the pre-pandemic and pandemic strain, the more abundant sequences with mutation frequency per genome less than a preset value, $f < d_v$, were randomly sampled and down-weighted by a coefficient, $D_w$, ranging from 5% to 50%. This procedure was done to balance the number of sequences sampled between the two strains. To obtain the average pairwise haplotype frequencies, $f_{ij}$, we repeated the resampling 200 times.

Next, we followed the procedure described above in Fig 1C–1F and calculated the two-way and three-way association UFE, Eqs 1 and 2, to infer the intra-protein network of interactions (Fig 2). To avoid divergence, we have considered only pairs when all four haplotypes were present in excess of a cutoff, $f_{ij} > 0.05$. The dependence of results on $(d_v, D_w)$, which infers between 15 and 22 compensatory sites, originates from the unequal presentation of the two strains in the database. We observe that slightly different weighting gives similar results within a plateau region in Fig 2F. We choose cases C, D, E based on the robustness with respect to the two sampling parameters seen as the broad plateau in the 2D diagram, see panels C, D, E. These cases correspond to a roughly equal amount of each strain. If one strain is strongly over-represented, the network disappears (Fig 2A and 2B). Below, we choose the network variant shown in Fig 2D as the "golden middle" of the set.

## A primary mutation and compensatory sites

We obtain that site 248 in NA represents the primary site connected to multiple compensatory sites (Fig 2D). Thus, our three-way method, tested in simulation and analytically, shows that the new strain that has outcompeted the old strain, only because it had a primary mutation 248 and many compensatory mutations. The network has the typical appearance of a fitness valley network observed, for example, in HIV for drug-resistant mutations.

One might ask whether these inferred mutations in Fig 2 are simply a driving mutation with hitchhiker mutations present in the invading strain. As easy to understand, in this case, our method would give zero signal instead of the network inferred (Fig 2). To make this fact obvious, we consider an extreme example, where the population represents a mix of the uniform old and uniform new strain. We focus on the part of the genome where the two strain differ, for example, 101100... and 010011..., respectively. These differences can be either due to driving or hitchhiking mutations. Note that, for any pair of these sites, only two haplotypes $f_{ij}$ out of the four are present. For example, for the first and second sites, we have haplotypes 10 and 01. Hence, when calculating correlation $UFE_{ij}$ and $UFE_{ij0}$ in Eqs 1 and 2, these pairs are excluded from analysis due to the cutoff condition, $f_{ij} > 0.05$. Thus, by design, our method

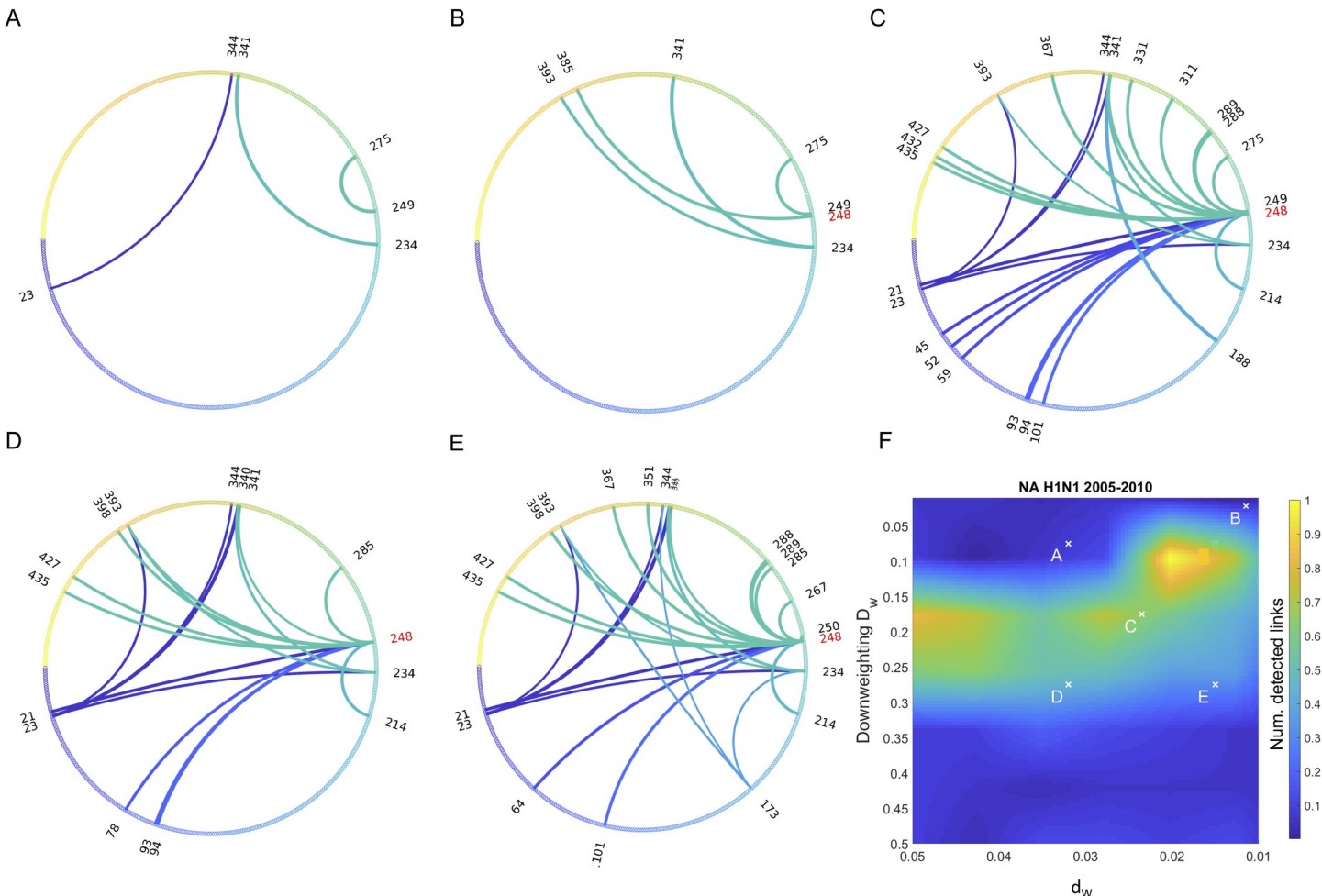

**Fig 2. Epistatic network predicted from sequence data on surface protein sequences of Influenza A H1N1 obtained between years 2005 and 2010.** The circular diagrams show the network of interaction between variable amino acid sites in the neuraminidase protein. Sequences with homology to the consensus less than $d_v$ were randomly re-sampled 200 times, with their number downweighted by coefficient $D_w$. (**F**) 2D heatmap showing the total number of links as a function of $d_v$ (X-axis) and $D_w$ (Y-axis). Different versions of wheels in **A-E** correspond to different choices of $D_w$ and $d_v$ shown by crosses in **F**. All links have estimated $E > 0.5$. (A-E) Colors correspond to different locations in the protein.

does not measure the difference between strains, but only a specific type of association between the fluctuations of alleles that is caused by epistasis.

## Structural interpretation

It is instructive to place the inferred epistatic sites on a three-dimensional protein structure (Fig 3). The active pocket of NA (purple) serves to bind sialic acid on target cell surface. We observe that the inferred primary mutation at residue 248 is located near the active pocket. The inferred compensatory mutations (Fig 2D) helping the mutant strain of NA to improve its fitness are all located on the protein surface in $\alpha$-helixes connecting and determining the mutual orientation of $\beta$-sheets. Inferred primary mutation 248 was previously shown to enhance the low-pH stability of NA [51]. It is ubiquitous in all influenza A H1N1 variants isolated after the 2009 pandemic, regardless of a geographic location [52–54].

Unlike the epistatic links, the false positives are not linked to any specific biology or proximity to 248. Linkage is indiscriminatory in this sense, as it is a simple consequence of

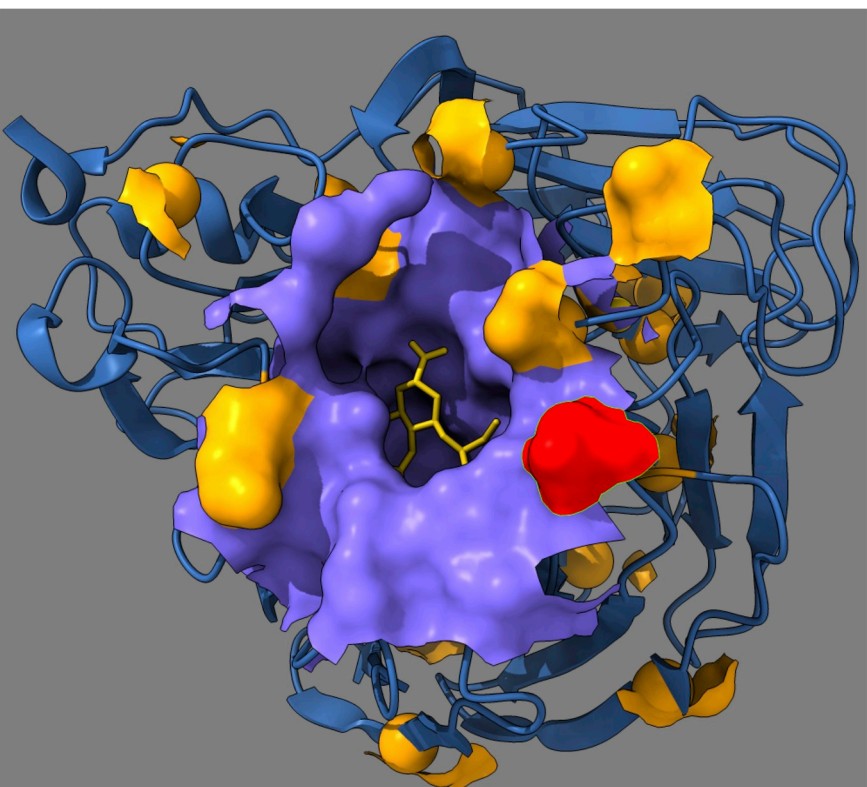

**Fig 3. Structural location of the predicted epistasis network for the neuraminidase of influenza virus.** The figure shows the three-dimensional structure of Influenza A H1N1 neuraminidase (PDB ID code 4QVZ). Colored spheres represent predicted epistatic residues from Fig 2D. Red sphere: Predicted primary mutation (residue 248 in Fig 2D). Orange spheres: Compensatory residues from Fig 2D.

stochastic phylogeny. Almost every pair of diverse sites in database is a potential false positive interaction, see Fig 1C as an illustration."

## Discussion

In the present work, we propose an efficient evolution-based method to tell apart co-variance caused by epistasis from co-variance caused by stochastic linkage effects due to common inheritance and indirect interactions. First, we average the observed haplotype frequencies over independent populations, then we select the links with a high co-variance, and then we apply a tri-way haplotype test for each candidate link to eliminate the residual false-positives. We validate the tri-way haplotype method using a simple analytical model (*Methods*) assuming a quasi-equilibrium state created by a slowly-moving traveling wave. The existence of quasi-equilibrium has been tested previously by simulation in a broad parameter range [46,50]. Intuitively, the distribution of alleles between sites has a sufficient time to attain the most probable state, i.e., the state with largest number of possible sequences given fitness.

To demonstrate the high fidelity of the method in a controlled environment, we used a simulated sequence set evolved in a Wright-Fisher population with a known epistatic network. In the case of a simple network topology and 40 loci, the method eliminated all false-positive interactions.

To illustrate the application of our method, we averaged haplotype frequencies over influenza H1N1 sequences obtained from a large number of geographic locations. We identified

primary and compensatory mutations responsible for the post-2009 strain. We did not address the origins or the history of the strain. We note that Influenza virus has been shown to map to the traveling wave theory [48,49], which justifies the use of our method assuming directional selection and the quasi-equilibrium assumption. Our results infer a single primary site and 15–20 strong compensatory mutations, which number is in the same general range as the number of compensatory mutations observed, for example, in drug-resistant strains of HIV. The inferred primary mutation, 248, has been observed in all influenza A H1N1 strains after the 2009 pandemic in various geographic locations [52–54]. It was shown to affect virus infectivity [51].

We did not find any empiric evidence in the literature for or against the inferred network of compensatory mutations. Hence, we make a new testable prediction for a future experimental test. Primary site 248 can be predicted by simple alignment and is well studied experimentally. Its compensatory sites, however, are impossible to detect *in vivo* without our method, due to strong linkage noise. It would be very useful to compare these predictions with the results of deep mutational scanning [55,56], which we hope will be available in the future.

It is worth noting that Influenza sequences represent a meta-population with many connected islands, not a single well-mixed population, nor completely independent populations. The approximation used in our work here is that the average over different geographic locations allows to obtain, at least, a partial average over the ensemble of independent populations. It is unclear to which extent the lack of the full ensemble average affects the estimate, but at least, we are able to compensate the residual linkage errors within that partial ensemble, something that has not been done before.

As compared to the existing techniques of elimination of indirect interactions, developed for different animal species that diverged millions of years ago [43,44], our method is designed for recently diverged populations (thousands of generations or less) of the same species and recently emerged mutations. Furthermore, our method is capable of eliminating stochastic linkage, which is of less importance when different species are compared. Where both methods can be potentially applied, such as calculating the fitness landscape of HIV Ab-binding regions [45], our method is much faster computationally, because it is local in the genome. Indeed, we can consider one pair of loci at a time without the need of simultaneous optimization of $L^2/2$ parameters of the full interaction matrix. Also, it helps to avoid the situation when the number of fitting parameter is too large, and the system is over-defined.

The main limitation of the proposed approach is that it assumes constant directional selection, as opposed to balancing selection or time-dependent selection, such as occurs under changing external conditions [57]. While the evolution of influenza in a population under the selection pressure of accumulating immune memory B cells has been mapped to the case with constant selection [48,49], the case of virus evolution under the CD8 T cell response [58] or the case of a virus co-evolving with its defective interference particle [59,60] have no such connection and require separate investigation.

Our aim here was to propose the first method that can, in principle, isolate linkage from epistasis. Our analytic derivation, under the stated conditions, demonstrates that the method works in a broad range. The future task will be to find the full region of the applicability of our method in terms of $N$, $s$, $U$, and the number of sites $L$. We hope to address the upper limit on $L$ in the future work.

The application of our technique to the metapopulation of influenza is based on the assumption that migration between local populations ("demes") is sufficiently slow, so that their evolution along stochastic trajectories remains mostly independent. The standard criterion of independence is that, for the genomic region of interest, the migration rate from directly connected demes is much smaller than the mutation rate per region. In the opposite limit when migration is very fast, the entire metapopulation is well-mixed, and the method

becomes useless. There exists a large intermediate interval of migration rates when the neighboring connected demes form well-mixed clusters, and the population represents a large number of such independent clusters, so the method still applies, except the linkage noise is increasing as their number is decreasing. When migration is so fast that a new best-fit genome arising in the metapopulation spreads faster to any deme than the local production of best-fit genomes by mutation, we have the panmixia case and the method ceases to apply.

To summarize, we proposed a technique to infer the epistatic effect on the evolution of locus pairs and tease it out from stochastic linkage effects. We hope that our approach and further development of this technique will prove useful for all researchers interested in finding fitness landscapes of various organisms from genetic samples.

## Methods

### Model

We simulate the evolution of a haploid asexual population of $N$ binary sequences. In an individual genome, each locus (site, nucleotide position, amino acid position) numbered $i = 1,2,...,L$ is occupied by one of two alleles, either the wild-type allele, denoted $a_i = 0$, or the mutant allele, $a_i = 1$. We use a discrete generation scheme in the absence of generation overlap (Wright-Fisher model). The evolutionary factors included in the model are random mutation with rate $\mu L$ per genome, constant directional selection, and random genetic drift due to random sampling of progeny. Selection includes an epistatic network with a set strength and topology. Recombination is absent. A previous modeling study shows that moderate levels of recombination can enhance epistatic detection [40]. We use the standard model of fitness landscape with pairwise interaction. The logarithm of the average progeny number of an individual genome, $W$, depends on sequence $[a_i]$, as given by

$$W[a_i] = -\sum_{i=1}^{L} s_i a_i + \sum_{i<j}^{L} s_{ij} a_i a_j \tag{3}$$

$$s_{ij} = E_{ij}(s_i + s_j)T_{ij} \tag{4}$$

where $T_{ij} = 0$ or $1$ is the binary matrix that shows interacting pairs. Here the selection coefficients $s_i$ and $s_j$ denote the individual fitness costs of two deleterious mutations that are partially compensated by each other. By the definition, $E_{ij}$ is the degree of compensation of deleterious alleles at sites $i$ and $j$. Values $E = 0$ and $1$ represent no epistasis and full compensation, respectively.

The formalism applies at negative values of $E$ as well, but we focus on positive $E$, which case is termed "diminishing returns epistasis". The reason for this choice is strong effects of epistasis and strong indirect interactions in this region. The analytic derivation in the *Methods* applies at any $E < 1$, i.e., below the full compensation point, which covers most basic types of epistasis.

In our simulation example in Fig 1, we consider a haploid population with the initial frequency of deleterious alleles, $f_0 = 0.45$, beneficial mutation rate, $U = 0.07$, fixed selection coefficient, $s = 0.1$, population of $N = 1000$ individuals, $L = 40$ sites, and fixed epistatic strength $E = 0.75$. The core Monte-Carlo simulation code for Fig 1 is written in MATLAB and deposited at site https://github.com/rbatorsky/hiv-recombination. It can be modified for different types of epistasis and recombination. The code for Fig 2 (data analysis) has been uploaded to https://github.com/irouzine/Pedruzzi.

### Main approximations

We assume that the system is under directed selection and in the multiple-mutation regime (traveling wave regime), which takes place if $\log(NU) \gg \log(s/U)$ [12]. In this case, we have

interfering selection sweeps occurring at many sites at once. We are far from mutation-selection-drift equilibrium, so that reverse (deleterious) mutations are negligible. In a broad parameter range, an adapting population can be represented by a slowly-moving, narrow peak in fitness coordinate [7,9,12,14,16]. Evolution is slow, because the limiting factor is the addition of a rare beneficial mutation established within a highly-fit genetic background [7,12]. Because the fitness distribution moves slowly, the entropy (the log number of possible sequences given fitness) of the mutation distribution over genomes has enough time to reach its current maximum, restricted by the current average fitness of the population. This situation is called "quasi-equilibrium". At each moment, each fitness class has enough time to reach the most probable, most chaotic state given its fitness. Previously, we verified the validity of quasi-equilibrium in a broad range of parameters and initial conditions after time $\sim 1/<s>$ [46].

## Linkage measure

We will use a binary measure of allelic correlations defined in Eq 1, where $f_{00}, f_{10}, f_{01}, f_{11}$ are the haplotype frequencies averaged over the ensemble of populations [46]. UFE performs similarly to more traditionally used measures, such as Lewontin's $D'$, Pearson correlation coefficient, $r^2$ [40], or mutual information [43,44]. As compared to these measures, UFE has the unique advantage of directly measuring the degree of mutual compensation of two alleles $E$, provided they do not interact with other sites. If a pair of loci does not interact with the other loci in the genome, we have $UFE = E$. If they are a part of a network, this measure overestimates $E$ [46]. In the main text, we calculate UFE for every pairs of sites (Fig 1A). We leave only those pairs where $UFE$ exceeds a set threshold of 0.6.

## Tri-way linkage measure

To test whether a detected correlation for a pair of sites $i,j$ is due to direct interaction rather than linkage or indirect correlation, we also calculate the three-way measure, Eq 2, where 0 in the third position selects only for the sequences with the consensus allele 0 at a chosen site connected to one site of the tested pair. We consider all possible connected sites as 0-nodes and calculate the minimum value of $UFE_{ij0}$ over all possible 0-nodes. Finding the minimum not only detects a detour but also finds the most important detour if there is more than one. Thus, we can identify and remove false-positive links as those with a low ratio $\min(UFE_{ij0})/UFE_{ij}$.

## Analytic test of the method

To demonstrate, in the general form, that the above method works on indirect interactions, we consider a simplified case of the fitness landscape model in Eqs 3 and 4. We assume a fixed selection coefficient $s_i = s_0$ and a fixed epistatic strength $E_{ij} = E$ (Eqs 3 and 4). Then, we can fully characterize a genome by the numbers of interacting allelic clusters of different size, $i$. Each such cluster comprises $i$ directly interacting alleles. Let $k_i$ denote the number of clusters with $i$ alleles and $b_i$ interactions. Then, from Eqs 3 and 4, we can express log fitness $W$ as a sum over clusters of different size (Fig 4)

$$W \equiv -s_0 f_0 L = -s_0 \sum_{i=1}^{i_{max}} k_i (i - 2Eb_i) \tag{5}$$

New notation $f_0$ has the meaning of the effective frequency of non-interacting alleles that would have the same total fitness, $W$. The number of interactions, $b_i$ at $i \geq 3$, depends on the topology of the epistatic network. Here $b_1 = 0$, $b_2 = 1$ for any topology. For the chosen network of double arches, we have clusters of single, double, and triple alleles, and $b_3 = 2$ (Fig 4).

**Fig 4. An epistatic network made of double arches used for simulation and analytic derivation.** Numbers $k_i$ denote the numbers of clusters with $i$ connected sites in a genome. Dots show deleterious alleles. Interactions between the existing deleterious alleles are shown in red.

As mentioned above, we assume quasi-equilibrium, as determined by the current fitness. At each moment of time, numbers $k_i$ are determined by the condition that the entropy of the system is maximal given its fitness, Eq 5. The evolving population reaches the maximum-entropy state at the given fitness level with respect to the polymorphic sites. Entropy $S$ is defined as the log number of possible sequence configurations (for example, 0111100101)

$$S = \log\left[\prod_{i=1}^{i_{max}} C_{L_i}^{k_i}(n_i)^{k_i}\right] \tag{6}$$

where $L_i$ is the number of all possible locations for a cluster of size $i$, and $n_i$ is the number of each cluster's configurations (shapes). The values of $L_i$ and $n_i$ depend on the network topology.

Previously, we applied this argument for several topologies to derive the numbers of clusters of different size [46]. We showed, for the topology in Fig 4, that the frequencies of clusters of size $i = 1, 2$ and 3, denoted $f_i = k_i/L$, are related as

$$f_2 = \frac{1}{3}f_1^{2-2E}, f_3 = \frac{2}{3}f_1^{3-4E} \quad E < \frac{3}{4} \tag{7}$$

The 1st and 3d site in each triplet in Fig 1B do not interact directly, but only indirectly through site 2. For these two sites, the haplotype frequencies are

$$f_{11} = 3f_3 + f_1^2, f_{10} = \frac{3}{2}f_2 + f_1 \tag{8}$$

When epistatic interaction is sufficiently strong, as given by the condition $E > 1/2$, triplets dominate numerically over single alleles and doubles, as given by $f_1 \ll f_2 \ll f_3$ (Eq 7). From Eqs 7 and 8, using these strong inequalities, we can approximate the haplotype frequencies as

$$f_{11} \approx 2f_1^{3-4E}, f_{10} \approx \frac{1}{2}f_1^{2-2E} \qquad \frac{1}{2} < E < \frac{3}{4} \tag{9}$$

Using covariance measure $UFE_{ij}$ defined in Eq 1, we obtain

$$UFE_{ij} \approx \frac{1}{4(1-E)} \tag{10}$$

For directly interacting sites 1 and 2 (Fig 1), we previously obtained $f_{11}^{dir} \approx f_1^{3-4E}, f_{10}^{dir} \approx \frac{1}{3}f_1^{2-2E}$ [see [46], Supplement, Eqs (3.29) and (3.30)]. This gives the same result, Eq 10. We observe that the indirect covariance between sites 1 and 3 is as strong as for directly interacting sites.

However, if we calculate three-way measure $UFE_{ij0}$ in Eq 2 instead of $UFE_{ij}$ by including only the sequences with majority allele 0 at site 2, then, instead of Eq 8 and Eq 9, we obtain

$$f_{101} \approx f_1^2, f_{100} \approx f_1 \tag{11}$$

$$UFE_{ij0} \approx 0.$$

Thus, the phantom covariance disappears when we select only the sequences with a majority

allele inserted between the two tested sites. This result is intuitively clear: by the definition of fitness (Eq 3), only minority alleles interact with each other, while majority alleles form a neutral background. The same method turns out to be extremely effective for eliminating false-positive interactions created by linkage (compare Fig 1B with Fig 1E).

We also repeated the same derivation for a more complex topology of closed squares (S1 Text and S1 Fig). The results are discussed in the main text.

## Sequence preparation

We have applied the three-way test to influenza virus. We performed a multiple progressive alignment for amino acid sequences of Neuraminidase protein of Influenza A virus strain H1N1 obtained from public database https://www.fludb.org. We focused on NA because of the massive amount of sequence data and because strong changes in NA are responsible for the higher infectivity of the pandemic strain.

The amount of data must be sufficiently high to ensure that each three-site haplotype entering UFE be represented by a large number of sequences, with their inverse square root being the relative error. In our case, this condition was fulfilled by setting cutoff at allelic frequency $f > 5\%$. We downloaded 8440 sequences of NA from a public database (https://www.fludb.org). They were collected worldwide, from different geographic locations, from year 2005 and year 2010, and included both pre-pandemic and post-pandemic strain. All sequences were aligned. We considered only the sites that were strongly polymorphic ($>5\%$) sometimes in the window of 5 years, during which time they contribute to the calculated correlation measures. Then, we found the common consensus (majority) allele for each amino acid position. Note that we have chosen the common consensus of the entire set as the reference sequence for calculating allele frequencies. The choice of the reference sequence does not matter for the site-site correlations and the inferred network.

Pairwise distances between sequences were computed using pairwise alignment. The obtained consensus, defined as the most frequent variant in the population, served as a universal reference to binarized data sequences. Before applying the detection algorithm, the protein sequences were binarized, by direct comparison of each sequence to the consensus. Each amino-acid residue was set to 0 or 1 for consensus or non–consensus. Although combining all amino acid variants per site ignores the specific biochemistry of substitutions, this approach greatly reduces the number of haplotype combinations and also increases the sensitivity by effectively increasing the haplotype frequencies.

Next, we measured the mutational frequency for each sequence along sequences and for each site across sequences. The subset of low-diversity sequences with allelic frequency below a cut-off $d_v$ was randomly sampled and down-weighted according to a set coefficient, $D_w$. Then, we determined the average pairwise and three-way haplotype frequencies for all pairs and triplets of sites, as described in the *Results* section.

## Supporting information

**S1 Text. Derivation of UFE for the closed square topology.**
(PDF)

**S1 Table. Direct and indirect UFE values for the square topology.** Index 0 indicates a 3-way measure.
(TIFF)

**S1 Fig. Square topology UFE calculation.** A) Possible configurations and their symmetry. B) Indirect interaction pairwise. C) Indirect interaction three-way, with a fixed zero at a node. d)

Indirect interaction with two fixed zeros. E) Direct interaction pairwise. F) Direct interaction three-way.
(TIFF)

**S2 Fig. Dependence of 5 different types of linkage measure (S1B–S1F Fig) on epistatic strength predicted for square topology.**
(TIFF)

## Acknowledgments

We thank Martin Weigt and Alessandra Carbone for useful comments.

## Author Contributions

**Conceptualization:** Igor M. Rouzine.

**Data curation:** Gabriele Pedruzzi.

**Formal analysis:** Igor M. Rouzine.

**Funding acquisition:** Igor M. Rouzine.

**Investigation:** Gabriele Pedruzzi, Igor M. Rouzine.

**Methodology:** Gabriele Pedruzzi, Igor M. Rouzine.

**Project administration:** Igor M. Rouzine.

**Software:** Gabriele Pedruzzi, Igor M. Rouzine.

**Supervision:** Igor M. Rouzine.

**Validation:** Gabriele Pedruzzi, Igor M. Rouzine.

**Visualization:** Gabriele Pedruzzi.

**Writing – original draft:** Gabriele Pedruzzi.

**Writing – review & editing:** Igor M. Rouzine.

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
