## [Decision Letter · Decision Letter 0]

7 Feb 2021

Dear Dr. Rouzine,

Thank you very much for submitting your manuscript "An evolution-based high-fidelity method of epistasis measurement: theory and application to influenza" for consideration at PLOS Pathogens. As with all papers reviewed by the journal, your manuscript was reviewed by members of the editorial board and by several independent reviewers. In light of the reviews (below this email), we would like to invite the resubmission of a significantly-revised version that takes into account the reviewers' comments.

We cannot make any decision about publication until we have seen the revised manuscript and your response to the reviewers' comments. Your revised manuscript is also likely to be sent to reviewers for further evaluation.

Sincerely,

Santiago F. Elena, PhD

Guest Editor

PLOS Pathogens

Raul Andino

Section Editor

PLOS Pathogens

Kasturi Haldar

Editor-in-Chief

PLOS Pathogens

orcid.org/0000-0001-5065-158X

Michael Malim

Editor-in-Chief

PLOS Pathogens

orcid.org/0000-0002-7699-2064

Dear Dr. Rouzine,

Thanks for giving us the opportunity to review your work, which is tackling a most relevant question in evolutionary virology. As you will see, the three reviewers find merits in your work but also detect some important weaknesses that need to be addressed before the manuscript can be accepted in PLoS Pathogens.

You will notice that two reviewers agree that you haven't provide strong enough evidence that your method works and which parameter regimes is expected to perform best. Also, an effort for clarity, specifically aiming to a readership of experimental virologists, must be done. Linking with this criticism, making available your code for others to use it is considered as essential.

All reviewers also point to many typos and places where the manuscript needs to be polished. Please, pay close attention to them.

Looking forward to see your revised version.

Santiago Elena

Reviewer's Responses to Questions

**Part I - Summary**

Reviewer #1: This study is valuable in its approach of using population genetics theory to devise a method of detecting epistasis among sequences sites. The approach is rigorous and well defended.

Reviewer #2: The authors propose a version of the linkage-disequilibrium test for detecting pairwise epistasis in sequence data. A major obstacle in detecting epistatic pairs is that many other pairs are also at linkage disequilibrium, due to hitchhiking or due to indirect interactions. The authors propose a procedure for reducing this noise which relies on estimating conditional LD, conditioned on a potential confounding site being at its wildtype state. They apply this approach to identifying epistatic networks in influenza H1N1 neuraminidase data and find that a previously known site 248 near the active pocket is involved in many interactions.

The authors ask an important question. Their core idea makes sense and is promising. However, I see two major issues with this manuscript, which I will discuss in detail below. The first problem is that the authors have not demonstrated how well their method works. In what regimes is it expected to work? When does to fail? The second problem is the organization of the manuscript and a lack of clarity in many places, which make reading it difficult and at times frustrating, even for a specialist.

Reviewer #3: attached

**Part II – Major Issues: Key Experiments Required for Acceptance**

Reviewer #1: (No Response)

Reviewer #2: 1. INSUFFICIENT EVIDENCE. The idea to detect epistasis using linkage disequilibrium is not new. It is clear why it works in sexual populations or in a system at mutation-selection balance. Here, the authors argue that essentially the same idea can also work in rapidly adapting asexual populations, such as the influenza virus. But why and when it works is unclear. Figure 2 convinces me that it works sometimes. But I think for this paper to have impact, the authors need to provide at least some guidance as to when the method works and when it fails.

For this method to work, the epistatically interacting sites must be at minimum simultaneously polymorphic. The triple-site method requires three sites to be polymorphic at the same time. The genetic diversity of a population depends on the population genetic parameters, as the authors are well aware. For example, if the population evolves in the successional mutation regime, the UFE statistic cannot be evaluated and this method will not work. But it is not clear to me whether simply being in the concurrent mutation regime is sufficient. The authors heavily rely on their earlier paper (Pedruzzi et al, PLoS Comp Biol 2018), but I could not find a clear delineation of different parameter regimes. Ideally, I would like to see a phase diagram in the U, N and s space for a some simple epistatic model like the “double arches” showing where the UFE statistic between a specific pair of sites correlates well with the true value of E, where it does not correlate and where it cannot be evaluated.

The second issue that I see is that the ability to detect epistasis using this method may depend on the structure of the epistatic network. Figure 2 shows that it works for the simplest possible structure, the “double arches”. We do not have a great picture of how epistatic networks look like within actual proteins. But I think evaluating the performance of this method for some more complex networks would be necessary.

The third issue that I am concerned about is how the amount of data and the sampling protocol might affect method’s performance. Most sites that carry beneficial mutations, e.g., in influenza, are polymorphic only transiently. This presumably means that the proposed test has to be applied to sequences sampled within a certain time window. So, how should one choose this window? And how sensitive is the method to this choice? I am guessing me that the downweighing that the authors did in their analysis of the influenza data set address this issue. But that was not clearly explained and seems ad hoc. In any case, this seems like an important issue and should be addressed directly in the paper. Also, how much data is actually needed in practice to reliably estimate the conditional UFE?

2. CLARITY. There are several major clarity issues. Minor issues are listed below.

I think the authors made a poor choice in how to introduce their approach. First, the second half of the intro where the authors essentially summarize part of their previous paper (Pedruzzi et al, 2018) is not understandable on its own without actually reading the supplement of that paper. Second, this theory is only relevant in so far as it provides sort of an explanation why the LD-based method might work in adapting asexual populations. But it is actually unnecessary for understanding the basic idea behind the UFE statistic, which is just a way to detect LD. Third, this explanation is not even satisfactory because it remains unclear when the assumptions made in their model actually hold (see above). So, the reader becomes very confused and skeptical before getting to the Results. Fourth, the UFE statistics are not introduced in the Results section. I think they should be introduced, since it is the central contribution of this methodological paper.

I found the analytical results for the double-arches model quite helpful. I think the authors should consider discussing these results in the main text.

The authors stress multiple times in the text that it is important to average the UEF statistic across multiple populations to reduce linkage noise. But they do not do this explicitly in their analysis of the flu data. This has to be addressed.

As mentioned above, the downsampling procedure is not part of their method, at least as described, and so its purpose has to be clearly explained.

Finally, I don’t quite understand how the authors estimate allele and haplotype frequencies from their data, given the fact that the sequences were sampled over multiple years. The authors appear to estimate frequencies from the entire alignment, which seems wrong because it does not represent a snapshot of a population. I also don’t understand how the authors are confident that the mutation at site 248 that they detected actually genetically interacts with other sites, given that this mutation is present in all post-2009 N1 variants? My understanding (but I may be wrong) is that pre-2009 H1N1 lineage was outcompeted by the 2009 “swine flu” variant. So, couldn’t it be then that the mutation at site 248 was simply present in the 2009 founder strain and is now shared by all of its descendants, without epistatically interacting with any other sites? What’s the evidence that it is actually epistatic?

Reviewer #3: attached

**Part III – Minor Issues: Editorial and Data Presentation Modifications**

Reviewer #1: p. 11 - The explanation for using the triple-site haplotype method should be made clearer at this point, even though it is explained in the Methods section.

p. 12 - Although it seems a reasonable assumption, there should be some justification for assuming an epistatic network for NA. Surely there is some empirical evidence for this.

p. 15 - Is there any empirical evidence that the 15-20 compensatory mutations are actually compensatory? Also, the identification of the primary site could presumably be made by simply comparing aligned sequences from before and after the pandemic. It would be useful to comment on this.

More generally, the method has been tested using compensatory epistasis, but is there any evidence it works with other forms of epistasis?

This is an editorial comment. There are minor errors of sentence structure or typographical errors throughout the manuscript. These seem typical of writers that are not native English speakers and could be easily corrected by a native speaker or by the journal.

Reviewer #2: (No Response)

Reviewer #3: attached

PLOS authors have the option to publish the peer review history of their article (what does this mean?). If published, this will include your full peer review and any attached files.

Reviewer #1: No

Reviewer #2: No

Reviewer #3: **Yes: **Christina L. Burch
---

## [Decision Letter · Decision Letter 1]

11 May 2021

Dear Dr. Rouzine,

Thank you very much for submitting your manuscript "An evolution-based high-fidelity method of epistasis measurement: theory and application to influenza" for consideration at PLOS Pathogens. As with all papers reviewed by the journal, your manuscript was reviewed by members of the editorial board and by several independent reviewers. The reviewers appreciated the attention to an important topic. Based on the reviews, we may consider this manuscript for publication, providing that you modify the manuscript according to the review recommendations.

Let us apologize for the long time it has taken to to reach a decision. I'm afraid to say that reviewers are still quite dispar in their opinions, from rejection to minor revision. After our own reading, like the three reviewers, we found your study very important and of interest for a very broad audience. However, we do concur with the most critical reviewer that the manuscript is still far from being clear enough for the (mostly) non-technical audience of PLoS Pathogens. Actually, some of us found PLoS Computational Biology a much better venue for this particular study. But this is just a personal preference.

Very rarely PLoS Pathogens gives second opportunities, however, given the interest of the manuscript. We'd like to consider a new largely revised version tackling the concerns by the most critical reviewer in a very constructive manner. not simply ignoring them. If you decide to do so, then we'll most likely reach a decision without sending out to a third round of reviewing.

Sorry we can't be more positive this time.

Sincerely,

Santiago F. Elena, PhD

Guest Editor

PLOS Pathogens

Raul Andino

Section Editor

PLOS Pathogens

Kasturi Haldar

Editor-in-Chief

PLOS Pathogens

orcid.org/0000-0001-5065-158X

Michael Malim

Editor-in-Chief

PLOS Pathogens

orcid.org/0000-0002-7699-2064

Dear Dr. Rouzine,

Let me apologize for the long time it has taken to to reach a decision. I'm afraid to say that reviewers are still quite dispar in their opinions, from rejection to minor revision. After my own reading, like the three reviewers, I found your study very important and of interest for a very broad audience. However, I do concur with the most critical reviewer that the manuscript is still far from being clear enough for the (mostly) non-technical audience of PLoS Pathogens. Actually, I'd found PLoS Computational Biology a much better venue for this particular study. But this is just my personal feeling.

Very rarely PLoS Pathogens gives second opportunities, however, given the interest of the manuscript. I'd like to consider a new largely revised version tackling the concerns by the most critical reviewer in a very constructive manner. not simply ignoring them. If you decide to do so, then I'll most likely reach a decision without sending out to a third round of reviewing.

Sorry I can't be more positive this time.

Reviewer Comments (if any, and for reference):

Reviewer's Responses to Questions

**Part I - Summary**

Reviewer #1: The authors appear to have done a thorough job of responding to the reviewers' comments.

Reviewer #2: The authors mostly addressed my concerns related to the structure of the paper. I found the current manuscript much more readable. The additional analytical work now provided in the supplement is also a step in the right direction. However, two major concerns remain. First, the present manuscript still does not give the reader a good sense of when the method works and how well, when it is expected to fail and how. Second, the claims regarding the influenza analysis remain unsubstantiated and the analysis remains unclear. In addition, my minor concerns from the first round of review have been apparently not conveyed to the authors. I reiterate several of them below.

Reviewer #3: This revised manuscript is greatly improved in terms of clarity. Most of the issues I raised have been resolved and I am most appreciative of the care taken to provide greater intuition for the methods choices. I now have only one major concern, one moderate suggestion, and a few copy editing suggestions. Once the major concern has been addressed, I think the manuscript will make a strong contribution to PLoS Pathogens.

**Part II – Major Issues: Key Experiments Required for Acceptance**

Reviewer #1: No major issues.

Reviewer #2: 1. ROBUSTNESS OF THE METHOD. PLoS Pathogens is a journal that is read by a fairly wide audience, including practitioners who might be interested in applying the method proposed here to other viruses. So, in my opinion, a paper published in this journal needs to go a bit beyond a mere “proof of principle”. Doing so is not actually too laborious.

Specifically, I think that a typical PLoS Pathogens reader would want to have at least some sense of what performance to expect of the method presented here. In fact, the author’s own analysis of influenza exposes the need for a sensitivity analysis. To address this concern, maybe the authors can use some quantitative metrics of performance, e.g., the rate of false positive and false negative epistatic pairs detected by their method, and report how their method performs with respect to these metrics as evolutionary parameters vary, at least in their double arches topology shown in Figure 1B. This is fairly straightforward to do. The authors know better than me which parameters are most important to vary, but I’d certainly be interested to know at least how the strength of epistasis E and the number of independent replicates affect the performance.

2. INFLUENZA CLAIMS. This part of the paper has the following two major issues.

2A. The description of the analysis of the influenza data remains unclear in several places, its organization could be streamlined, and some key pieces of information are missing. All of this makes it very difficult to understand what has been done, and makes this analysis not reproducible as described.

I think the following two specific points need to be addresses before I’d be comfortable recommending this paper for publication.

2A-i. Describe the analysis in more detail, provide key pieces of information in the text, make data and code available, as described in (a) through (h) below.

(a) The authors say that the sequences were split into several subsets by geographic location (LL 281-287). Into how many geographic groups have the authors actually split the data? How coarse-grained are these locations? How many sequences of each type (pre-pandemic and pandemic) are in each location? As far as I can tell, this information is currently not provided anywhere. Please provide it, e.g., as a data table, and report some key summary statistics in the text.

(b) Was the frequency cutoff of 5% applied to each location separately or is it 5% of the total number of sequences, i.e., 8440 * 0.05 = 422? I suspect it is the latter. In this case, please state this explicitly. But then I wonder how is the UFE statistic evaluated for sites that are not polymorphic in some sub-populations? Or does this never happen? Please discuss this.

(c) Please provide the estimated consensus sequence as well as the binarized individual sequences as a supplementary data files. Or at the very least provide the code and the raw sequence data file that you used that would generate all these intermediate data files.

(d) Please provide the estimated individual allele frequencies, bi-allele frequencies and the conditional allele frequencies for each geographic location.

(e) Please show the distribution of mutant allele frequencies as a figure and please provide an estimate of the fraction of sequences around each peak.

(f) I also would very much like to see a phylogeny showing how the pre-pandemic and pandemic clades are related to each other. Specifically, I would like to see how many mutations separate the pandemic clade from the rest of the pre-pandemic variants. This information is critical for understanding how the authors arrive at their epistasis network for site 248 later on.

(g) Rather than saying “multiple times”, please specify how many times exactly has the resampling been performed (L 314).

(h) Please provide the estimated values of the UFE and conditional UFE statistics for each location for each pair of sites in a supplementary data file.

2A-ii. Streamline this part of the paper.

LL 281-287. I don’t understand what the purpose of this paragraph is. Is it a description of what is being done or an introduction into what will be described next? The next paragraph describes the processing of the data, so I infer that the current paragraph is introductory. Then it is unnecessary and confusing. The previous paragraph gives enough background information, and diving right into a brief description of what is being done would be preferable. The current paragraph also mentions how sequences were divided among geographic locations, which is not explained in more details anywhere else, as far as I could tell. I suggest removing the redundant parts of the paragraph and providing missing information in the Methods section and as a supplementary data, as suggested above.

LL 288-300. IMO, this paragraph largely belongs to the Methods section and also requires more details, as discussed above.

2B. There is currently not enough support for the main result that the mutation at site 248 genetically interacts with other sites. I once again ask the authors to explain how they reject the following hypothesis. Suppose that all pandemic N1 variants carry 15 to 22 mutations (including a mutation at site 248) that differentiate all these variants from all pre-pandemic variants. In this case, there is simply no information about the epistatic interactions between these 15 to 22 sites. All these mutations could be hitchhikers. We only know that they form a single haplotype which is sweeping to fixation. The argument that “our three-way method, tested in simulation and analytically, shows…” is not sufficient because neither the simulations nor the analytical calculations have explored a scenario where a new strain with dozens of new mutations invades a population. This is not a quasi-equilibrium travelling wave scenario that was at least to some extent explored in the first part of the paper. So, the fact that the conditional UFE method reveals some significant pairs in this analysis is not sufficient to convince me that site 248 is actually involved in any epistatic interactions, let alone the claim that “the new strain that has outcompeted the old strain, only because it had a primary mutation 248 and many compensatory mutations.”

A smaller but still important issue is that the phase diagram in Figure 4F shows that the number of detected interacting site pairs is quite sensitive to the choice of the resampling parameters. But the authors do not specify how these parameters should be chosen. They simply declare by fiat that choices A and B are incorrect whereas choices C, D, E are correct. I don’t understand what justifies this choice.

Reviewer #3: Attached

**Part III – Minor Issues: Editorial and Data Presentation Modifications**

Reviewer #1: The authors, in their response to my comment, state:

(p. 12 - Although it seems a reasonable assumption, there should be some justification for assuming an epistatic network for NA.)

“We do not assume. We infer it. We detect true interactions and measure their magnitude. I made it clear now:”

Epistasis is defined as interactions among alleles that affect the phenotype. You are inferring associations among alleles, not epistasis. Also, you do not “detect true interactions” if you are inferring them. It would be more convincing if you could provide independent evidence of epistasis. This is a theoretical proof-of-concept after all.

The authors state on p. 6:

“Co-variation due to random linkage completely masks the epistasis signature in a population. The only way to resolve this issue is to average the haplotype frequencies over many independent populations with similar parameters under similar conditions. Without sampling multiple populations, it is not possible to infer epistasis in principle, due to the stochastic nature of phylogenetic relation of sequences. This fundamental limitation cannot be resolved by any existing or future method.”

Why can’t this limitation be resolved by any existing or future method? Why can’t multiple independent populations be sampled?

Reviewer #2: Throughout the text. It is stylistically preferable to avoid loose jargon such as “detour”, “kills indirect interactions” and others. “Detour” is used particularly often. I suggest that the authors either define what they mean by “detour” more precisely or simply replace it with something more precise like “indirect interaction” or “spurious interaction”.

LL 190. How is the UFE_{ij} statistic (or the UFE_{ij0}) evaluated if either f_{01} or f_{10} are zero?

LL 237-248. Please provide in the Results section the expressions for the UFE_{ij} and UFE_{ij0} statistics for epistatic and non-epistatic pairs obtained in the analytical calculations. I think it’s fine to relegate the calculations themselves to the Methods section. But the reader needs to see some substantiation for the statements made in the paragraph.

LL 245. It seems that calling this type of epistasis “diminishing returns” is not quite accurate (“diminishing returns epistasis” usually refers to beneficial mutations), and adding this label doesn’t add any value to this discussion. I suggest removing it.

LL 357-359. This statement “We note that Influenza virus has been shown to map to the traveling wave theory …” is somewhat misleading. Yes, antigenic drift in influenza is reasonably well described by the travelling wave theory. However, this is not the scenario that the authors tackle here. The pandemic 2009 flu is a relatively distant strain of H1N1 that invaded the resident H1N1 strain. So, it’s not just a single travelling wave situation, as mentioned above. Please rephrase.

L 427. Should there be a + sign in front of the second sum? Otherwise, I do not see how E = 1 represents compensation.

L 441. What exactly is f0?

L 441. Do U and µL represent the same quantity?

L 446. What does “epistatically isolated” mean?

LL 454-457. I think this statement requires some qualifications. I think the authors mean that the evolving population reaches the maximum-entropy state at the given fitness level with respect to the polymorphic sites. There could be other genotypes that have the same level of fitness but would requires flipping some sites that are fixed in the population.

Section “Analytic test of the method” (LL 479-523). I am confused. Are these calculations for the “double arches” topology shown in Figure 1 or for the “triple arches” topology shown in Figure 4. These paragraphs refer to both figures. See my related comment to Figure 4 below. Please resolve this confusion.

L 489. “interaction s” remove the space.

LL 510-511. I still don’t follow how the expression for UFEij is derived from Eq (9). It seems some approximations are made. Please state them explicitly.

LL 513-514. Please provide an expression for the UFE for a truly epistatic pair so that the expressions for UFE for epistatic and non-epistatic pairs can be directly compared.

LL 517-518. And the same for the UFE_{ij0}. Direct comparison of conditional UFE for both types of pairs would also be helpful.

L 541-544: What is a guide tree and how is it used? It seems that it is used to generate the consensus sequence, which is then used to binarize each amino acid residue in each sequence. But how the consensus sequence is obtained is not clear from this description.

Figure 4.

(1) This looks like “triple arches” to me, not “double arches”, so it is a different topology than the one that the authors explore numerically in Figure 1. I pointed this out in my first review, but this has not been addressed. Please resolve this.

(2) What does “active interaction” mean?

Reviewer #3: Attached

PLOS authors have the option to publish the peer review history of their article (what does this mean?). If published, this will include your full peer review and any attached files.

Reviewer #1: No

Reviewer #2: No

Reviewer #3: No

Figure Files:

Data Requirements:

Reproducibility:

References:

---

## [Editor Report · Decision Letter 2]

25 May 2021

Dear Dr. Rouzine,

We are pleased to inform you that your manuscript 'An evolution-based high-fidelity method of epistasis measurement: theory and application to influenza' has been provisionally accepted for publication in PLOS Pathogens.

Best regards,

Santiago F. Elena, PhD

Guest Editor

PLOS Pathogens

Raul Andino

Section Editor

PLOS Pathogens

Kasturi Haldar

Editor-in-Chief

PLOS Pathogens

orcid.org/0000-0001-5065-158X

Michael Malim

Editor-in-Chief

PLOS Pathogens

orcid.org/0000-0002-7699-2064

Dear Dr. Rouzine,

Thank you for the time and effort put into responding to the reviewers comments in a positive manner. I find this revised version much improved and I glad to recommend its acceptation. I'm sure this will be a very influential contribution to the field.

Best regards,
---

## [Editor Report · Acceptance letter]

15 Jun 2021

Dear Dr. Rouzine,

We are delighted to inform you that your manuscript, "An evolution-based high-fidelity method of epistasis measurement: theory and application to influenza," has been formally accepted for publication in PLOS Pathogens.

Best regards,

Kasturi Haldar

Editor-in-Chief

PLOS Pathogens

orcid.org/0000-0001-5065-158X

Michael Malim

Editor-in-Chief

PLOS Pathogens

orcid.org/0000-0002-7699-2064